# The Visualization of ISO/IEC29110 on SCRUM under EPF Composer

**Kittitouch Suteeca \* and Sakgasit Ramingwong \***

Department of Computer Engineering, Faculty of Engineering, Chiang Mai University, Chiang Mai 50200, Thailand
* Correspondence: kittitouch.s@cmu.ac.th (K.S.); sakgasit@eng.cmu.ac.th (S.R.); Tel.: +66-81-883-1386 (K.S.)

**Abstract:** In the midst of an increasingly competitive software industry, very small entities (VSEs) have inevitably faced many challenges. High user expectations, frequent changes of user requirements, and the need for rapid deployment are classic examples of these challenges. Many software companies attempt to implement measures for preventing or solving the aforementioned problems. The use of agile methodologies and the implementation of software development standards are usually perceived to be promising solutions to improve the quality of the software development process. Nevertheless, there are several strong incompatibilities between standards and the Agile approach to software development. For example, the need identified in the standards to create many quality artifacts does not conform to agility philosophies. Since Agile focuses on the working software over the documentation, the use of the Agile with standards can be difficult to implement. Additionally, there has been none guidelines for VSE therefore, an external consultant is usually required. This research analyzes various cases of implementing ISO/IEC29110, a software development standard developed especially for VSEs in Scrum environments. The results of this study provide an Eclipse Process Framework (EPF) for effectively and conveniently implementing this standard in Scrum software development.

**Keywords:** ISO/IEC29110; Scrum; software process improvement; Eclipse Process Framework

## 1. Introduction

There is a variety of software development methodologies. Different methodologies are appropriate for certain scenarios in software projects. According to a survey [1], the rate of successful software projects is approximately 60%. This is due to many challenges and problems, such as increasingly high expectations concerning the quality of software, continuous changes in user requirements, and the rapid evolution of technology. All of these make software development significantly more complicated than in the past. Many problems can occur in a software project. Software errors, requirement definition, and communication are found to be the most common causes of problems in development [2].

Agile software development is a philosophy describing a set of principles defined in the Agile Manifesto [3]. There are four elements in the agile principle: (1) Individuals and interactions over processes and tools; (2) working software over comprehensive documentation; (3) customer collaboration over contract negotiation; (4) responding to change over following a plan. The advantage of implementing the agile approach in a software project is that the process turns its focus to rapid and frequent delivery, as well as becoming more responsive to change. There are many models of agile methodologies such as Scrum, extreme programming (XP), Kanban, lean, etc. Scrum is currently one of the most commonly selected development methodologies in the industry [3].

Software development standards are sets of mandatory requirements developed and maintained by the agreement of professional groups or standards organizations. They are used as a framework to provide practices and methods in software development. In software engineering, there are various standards and management practices available for adoption.

In the software industry, several international ISO standards and standards from professional organizations such as Institute of Electrical and Electronics Engineer (IEEE) and Capability Maturity Model Integration (CMMI) are regularly implemented. The general objective of this is to enhance the quality of software development [4]. In a software project, the challenge of implementing standards is an increased level of sophistication. This requires additional development efforts. Modern development models such as Scrum are relatively incompatible with established standards for software development. Although the traditional standards for software development focus on the supported artifact and ensuring the documentation of each process [5], Scrum and other agile models focus on rapidly delivering the software product to a user with fewer supporting documents [6]. From the study of Kuhrmann et al., micro and small size companies only occasionally implement standards in their projects. The result showed that only 15.9% of the surveyed companies had implemented a Software Process Improvement (SPI) program based on standardized approaches like ISO/IEC15504 or CMMI [7].

Various cases of implementing ISO/IEC29110 in Scrum software development are studied in this paper. ISO/IEC29110 is a standard for lightweight software development, designed particularly for very small software entities (VSEs). VSEs are corporate organizations or development teams of entities with less than or equivalent to 25 employees. ISO/IEC29110 provides a reference for the processes of project management (PM) and software implementation (SI). In this paper, Eclipse Process Framework (EPF) Composer, an open-source content management tool, is used as the primary tool. This tool efficiently visualizes the ISO/IEC29110 standard process and the Scrum methodology. The results of our analysis provide a framework for effectively implementing the standards and the Scrum methodology for VSEs. This framework can be further adopted by organizations which plan to implement ISO/IEC29110 for their Scrum software development. This study focused on analyses the compatibility between standard and scrum methodology. The results are illustrating on EPF. The VSE who adopt the ISO/IEC29110 and scrum can use this framework as a project plan guideline.

This article is an analysis of compatibility between a software development standard and an Agile methodology on the management tool. The following sections are structured as follows: First, an introduction is given to the common problem in the software development organization. The scrum and the software development standard are the basic methodologies to prevent the problem. The second section explains the background of scrum methodology, ISO/IEC29110 software process reference model, and Eclipse Process Framework with the related work. The methodology for implement the framework on EPF is described in this section. In Section 3, the results of the analysis and the tailored process by the scrum and ISO/IEC29110 are provided. The EPF is a primary tool for display the tailored process. The discussion of the results, the case study, and future plan are in Sections 4 and 5 concludes the article.

## 2. Backgrounds

### 2.1. Agile Software Development and Scrum

The agile methodology is a technique that is often chosen for increased quality and rapid feedback, which support the ability to change directions when adjustments need to be made [8]. Communication and collaboration are important to agility. Fast reactions to change and continuous improvements are essential in agile development. Moreover, fewer artifacts are implemented in this development process. Scrum is a software development model with an iterative and incremental structure. Scrum's typical approach is to increase iterative progress and drive the team to achieve the target. The Scrum methodology is illustrated in Figure 1. Scrum begins with a number of product backlogs, which are described by the customers' requirements. In the next cycle, the Scrum team selects a set product backlog, called the sprint backlog, to develop. Sprint is a term for the period of production time, which varies among different Scrum teams and scenarios. A daily Scrum meeting is orchestrated every day in the morning in order to track and report

progress. In the end, before delivering a product to the consumer, the team must evaluate the achievement of all backlogs.

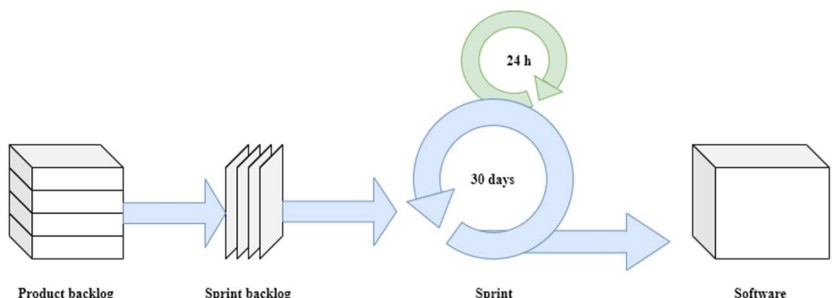

**Figure 1.** The Scrum process.

### 2.1.1. The Elements of Scrum

Scrum is the most common Agile methodologies used by respondents' organizations [9]. In a project, scrum techniques can be applied into the planning as well as other stages of development. There are seven main elements to consider for implementing the scrum on the software project.

- The product and sprint backlog: The product backlog is a list of new requirements, changes in requirements, bug fixes, or other tasks that the team is required to implement and deliver to the requirement owner. The sprint backlog is a list of selected backlogs that are to be implemented in each development sprint.
- The iteration length: The iteration length is a period of time in a sprint. The software team can schedule the duration according to the number of weeks, days, or hours.
- The sprint planning: The event at the beginning of each sprint to choose the product backlog items needed for producing the product. The product owner and the Scrum team discuss and prioritize the product backlog. The team is assigned a selection of high backlog items.
- The daily scrum: The daily scrum is a scheduled daily stand-up meeting for each development day. Meeting, discussion, feature development, or analysis may be included in this task.
- The sprint review: The sprint Review is an activity designed to review the completed sprint backlogs. This is generally performed on the last day of the sprint or at the start of the next sprint. Several issues which arose in the project will be identified and discussed for future mitigation strategies.
- The sprint retrospective: The sprint retrospective is the sprint's final event. This is the time in which the Scrum team considers what might be done better in an upcoming project. The discussion defines the opportunity to improve the product implementation and develop problem prevention methods.
- The sprint: The period of time in which to produce the product, following the sprint planning.

### 2.1.2. The Scrum Roles

Scrum Master, Product Owner, and Scrum team are the three positions that make up a project team in Scrum. All roles are required for the Scrum elements to run smoothly. The teams are self-controlling, and they typically consist of people with various professional backgrounds [10]. The description of the roles is as follows:

- Scrum master: The person who leads the team and sets the goals. This person is responsible for observing whether the team complies with the rules and completely understands the Scrum system.
- Product owner: The person who serves as an interface between the team and other participating parties (stakeholders).

- Scrum team: A development team with the aim of producing the product and achieving the target.

### 2.2. ISO/IEC 29110 Software Development Standard

The major constraint of a small software company is the limited number of development resources. It is undeniable that quality processes require extra resources and dedication. This makes the implementation of the large number of software development standards, e.g., ISO/IEC12207 or CMMI standards, significantly difficult. In 2010, Working group 24 from the ISO developed a standard in the field of software engineering for small software units or companies, with the aim of resolving such problems.

As a result, a lightweight process reference model (PRM) for a small software business, ISO/IEC29110: Software Process Lifecycle Standard for VSEs (Very Small Entities), was introduced. As mentioned, VSEs include business organizations or development teams which have between one and 25 members. ISO/IEC29110 is a standard for software development that responds to the characteristics and needs of VSEs. In this standard, two key processes are defined: project management (PM) and implementation of software (SI) [11]. The features of these two processes are as follows:

- The project management process aims to establish in a systematic way the activities involved in software implementation which enable the project's objectives to be achieved within the expected quality, time, and costs.
- The software implementation process aims to develop the software through the systematic performance of six activities: initiation, analysis, design, construction, testing, and delivery of new or modified software products according to the customer's requirements.

Five parts are included in the structure of ISO/IEC29110. The terms common to the standard series of ISO/IEC 29110 are specified in the first part. This includes the concepts, terms, and structure of the standard. The second part, Framework and Taxonomy, describes the concept and mechanism of constructing the VSE profile. Four profiles—entry, basic, intermediate, and advanced—are established in the standard. The entry profile requires the lowest criteria of entry, and the highest criteria are for the advanced profile. The auditor's appraisal guidelines and the specifications for certification are set out in the third part. This section of the standard is used to conduct VSE evaluations by certified auditors. The specifications for all profiles in this standard are defined in part 4 of the standard. This section presents the mapping to the ISO/IEC12207 source standard. The Management and Engineering Guide for VSEs is specified in the last section. As a guideline for implementation, it contains the definition of the method, mission, function, and work product.

The key of the process assessment is the process performance indicator. This indicator is defined in the third part of ISO/IEC29110 standard [12]. The process assessment framework is a compilation of the profile specification (ISO/IEC29110 part-4) and process performance indicator. The process assessment framework is presented in Figure 2. There are fifteen outcomes for PM process and thirteen outcomes for SI process. The outcome is criteria to measure the achievement of the implemented process. The outcomes of PM and IS are shown in Table 1.

### 2.3. Eclipse Process Framework Composer

The Eclipse Process Framework (EPF) Composer is a cost-free, open-source tool for the implementation, deployment, and maintenance of the process of an enterprise or individual projects by company designers, administrators, process engineers, team leaders, and project managers [9]. It provides process-engineering capabilities for individual projects to select, tailor, and rapidly assemble processes. A list of pre-defined processes can be generated on an EPF by the engineer. The process library includes the process's metadata such as Task, Role, Guidance, and Work products. In adjusting and changing the process for each VSE, which is needed to implement the ISO/IEC 29110 standard, the efficient use of the

composer program is helpful. The output of the EPF composer is an implemented process
that can be published on the website. An example of the EPF composer interface is shown
in Figure 3.

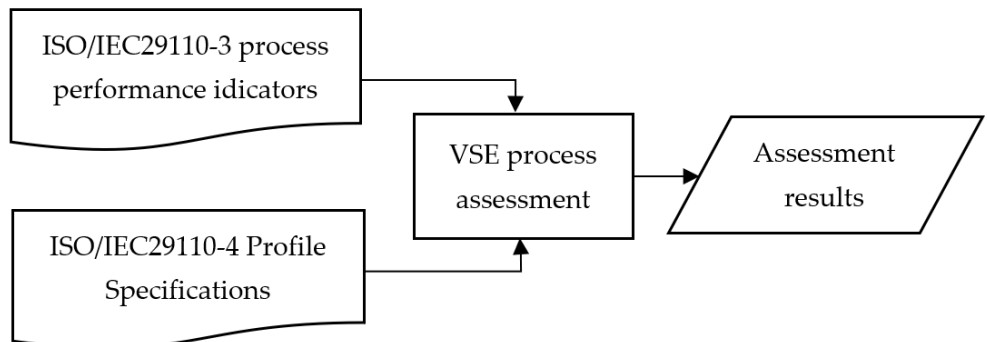

**Figure 2.** ISO/IEC29110 process assessment framework.

**Table 1.** ISO/IEC29110 process outcomes.

| Process Name | Project Management |
|---|---|
| Process outcomes | (a) the scope of the work for the project shall be defined;<br>(b) the tasks and resources necessary to complete the work shall be estimated (schedule, effort, cost, duration);<br>(c) planning for the execution of the project shall be developed according to the scope and the tasks defined;<br>(d) a software version control strategy shall be developed;<br>(e) planning shall be reviewed and agreed by the customer;<br>(f) progress of the project against the planning shall be monitored and reported;<br>(g) risks shall be identified and monitored during the conduct of the project;<br>(h) changes shall be addressed, analyzed and evaluated for cost, schedule and technical impact;<br>(i) relevant items of software configuration shall be identified and controlled including their storage, baseline, handling, and modifications;<br>(j) releases of items shall be controlled and made available to relevant stakeholders;<br>(k) product shall be completed and delivered to the customer as planning;<br>(l) meetings with the work team and the customer shall be held to guarantee that work done complies with the project requirements and planning;<br>(m) agreements resulting from meetings shall be registered and tracked;<br>(n) actions to correct planning problems and unachieved targets (schedule, effort, cost, duration) shall be taken;<br>(o) project closure shall be performed to get customer acceptance; |

**Table 1.** *Cont.*

| Process name | Software implementation |
|---|---|
| Process outcomes | (a) software requirements shall be defined;<br>(b) software requirements shall be analyzed for correctness and testability;<br>(c) software requirements shall be agreed by the customer;<br>(d) software requirements shall be baselined and communicated to work team and customer;<br>(e) software architectural and detailed design shall be developed and baselined;<br>(f) software architectural and detailed design shall describe the software<br>components and their internal and external interfaces;<br>(g) software components defined by the detailed design shall be produced;<br>(h) unit test shall be performed to verify the consistency with requirements and the detailed design;<br>(i) software shall be produced by integrating software components;<br>(j) software shall be tested and verified, the results shall be recorded;<br>(k) consistency and traceability between software requirements, software<br>architectural, software detailed design and software components shall be<br>established;<br>(l) defects identified in reviews, traceability analysis, tests and verifications shall be corrected;<br>(m) software configuration shall be integrated, baselined and stored in the project repository; |

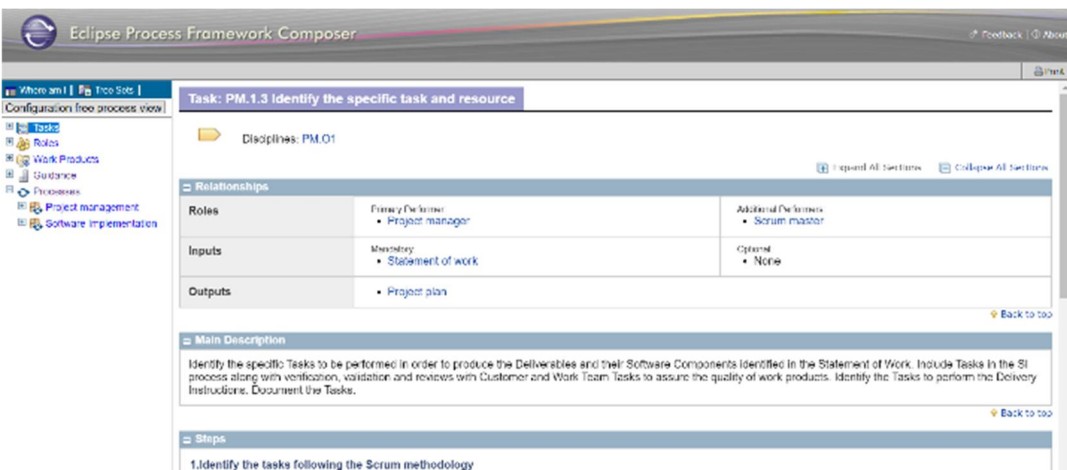

**Figure 3.** Example of the Eclipse Process Framework (EPF) Composer interface.

*2.4. Process Modeling*

To visualize the profile of the ISO/IEC 29110 standard with high quality and coherence, Buchalcevova, defined a methodology for the management and publication of the standard [13]. The high quality and coherence of the ISO/IEC 29110 profile was developed and described. The primary concept in the EPF composer is the classification of reusable main method content based on its application in processes. Almost all of the EPF Composer's concepts are categorized according to this separation. Method content describes the item to be created, the important skills required and provides sequence explanations describing how specific development goals are achieved. Processes assign the method con-

tent elements and relate them to semi-ordered sequences, which can be modified according to the VSE project.

The main data, using work products, roles, tasks, and guidance, constitute the method content. It is also possible to define guidance, such as through checklists, guidelines, or definitions, to provide the specifications for a process. Activities, delivery processes, and capability patterns are the elements used to represent a process in EPF. An activity can be nested to define breakdown structures, and activities can be related to each other in order to define a process as the main process element. Activities also include data referred to based on method content, e.g., roles, work products, etc. Activities are used to define processes. Within the EPF Composer, two key categories of procedures are supported, i.e., delivery process and capability pattern. The configuration of the EPF process library is shown in Figure 4.

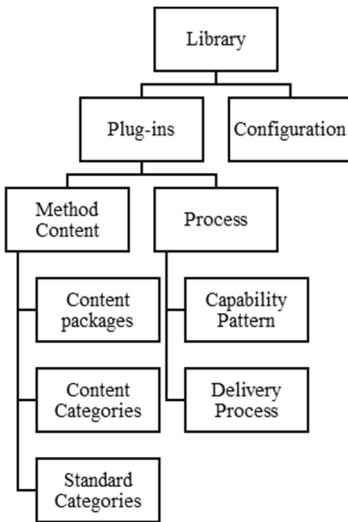

**Figure 4.** The structure of a process library from EPF.

### 2.4.1. Profile Structure

In the EPF Composer, the content structure requires both categories and pages. A category consists of methodological content, in which other categories and/or pages are included. Categories are used to separate the elements into logical groups or to encompass all the content. An example of the profile structure is shown in Figure 5. For this work, the profile structure used the basic profiles of the ISO/IEC 29110. The Scrum methodology compiles each form of content—roles, work products, tasks, and guidance—into the profile through mapping analysis. The guidance contains the key for process assessment as an outcome of the process.

### 2.4.2. EPF Composer Usage Guidelines

The composer is similar to the programming tool Eclipse IDE for JAVA. The library structure of a profile consists of the same class and method pattern. Class and method attributes are allocated by writing data into the field of each element. In the library, the method will inherit the author element.

As can be seen in Table 2, the use element in the EPF composer for the research will be chosen as the appropriate element. The compulsory element requires data to establish the content and the process of the system in the library. The role, tasks, work product, guidance, and disciplines are compulsory elements in the method content. Capability pattern and delivery process are the processes designed in the library. The optional elements are additional elements for the library. The element category helps the author to identify the kind of content in the content package.

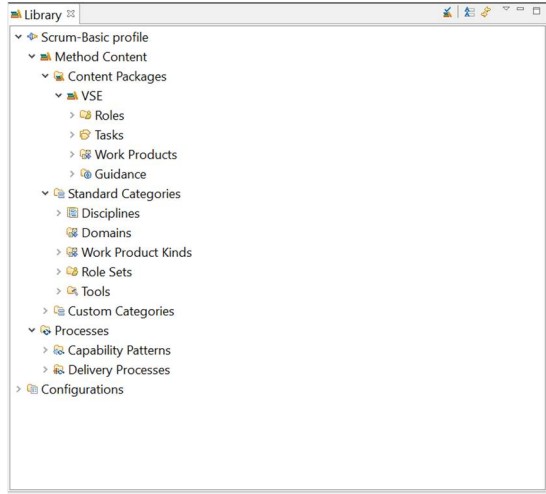

**Figure 5.** Example of a profile on EPF Composer.

**Table 2.** EPF Composer usage guidelines.

| Element in EPF | Mandatory | Optional |
|---|:---:|:---:|
| **Content package** | | |
| Roles | X | |
| Tasks | X | |
| Work product | X | |
| Guidance | X | |
| **Standard category** | | |
| Discipline | X | |
| Domain | | X |
| Work product kind | | X |
| Role set | | X |
| Tool | | X |
| **Custom categories** | | |
| Software tool | | X |
| **Processes** | | |
| Capability patterns | X | |
| Delivery Process | X | |

## 3. Results

Process modeling focuses on capturing and representing some reflecting angles of the system. Consequently, the model will create from the ISO/IEC29110 framework and Scrum methodology. The profile element mapping is the designed method for the model initiation. The provided elements in EPF are mapped to the ISO/IEC29110 and the scrum elements (Section 3.1). The analysis of the compatibility of standard and scrum is defining in Section 3.2. The basic profile is selected case from the standard. The roles and activities have been analyzing by the authors. The EPF is the tool for architects the new framework from analyzed data. The last, the data from modeling e.g., method content, capability pattern, task description, and process diagram are the results of this study.

### 3.1. Profile Element Mapping

The pre-defined category in EPF composer provides the basic elements for the architecture of the process. The categories for the EPF Composer should be implemented first. The pre-defined category was allocated to profile elements such as activity, work product, process, product, role, software tool, role sets, status, and task. The process assessment of ISO/IEC29110 can be applied using the ISO/IEC33004 process assessment model (PAM) [14]. The model consists of the achievement level of tasks and processes. There

are four levels of achievement—completely achieved, largely achieved, partially achieved, and not achieved. Completely achieved is the highest level of achievement. There is a systematic definition of complete or almost complete within the process. Not achieved is the lowest level. There is no evidence or related task in this process. The Scrum technique was applied to the ISO/IEC29110 framework in the case study, and ISO/IEC33004 provided a measure for determining the process's achievement [15].

The EPF composer provides the category of the content required to develop the process. In this study, the selected content categories were implemented for the framework. Table 3 shows the mapping of individual elements of the VSE profile to the recommended EPF Composer content categories, using the concrete example of the elements of the basic profile. In an example, the objective of the process is mapped to the disciplines category of EPF e.g., PM.O1: "The Project Plan for the execution of the project is developed according to the Statement of Work and reviewed and accepted by the Customer. The Tasks and Resources necessary to complete the work are sized and estimated".

**Table 3.** Mapping of individual elements of the very small entity (VSE) profile.

| Profile Element | Chosen EPF Composer Content Category | Basic Profile Example Element |
| --- | --- | --- |
| Objective | Disciplines | PM.O1 |
| Activity | Capability pattern | PM.1 Project planning |
| Work product | Work product | Statement of the work |
| Role | Role | Project manager |
| Outcome | Checklist | PM. Outcome (a) |
| Process | Delivery Process | Project management |
| Software Tool | Tool | Trello |
| Task | Task | PM.1.1 Get approval from the user |
| Work product status | Work product State | Baseline |
| Sub-task | Step | Get the approval (sign-off) |

Trello: The task management tool. Available in https://trello.com (accessed on 27 April 2021).

### 3.2. Analysis of ISO/IEC29110 Standard and Scrum

The study was conducted based on the job descriptions and job specifications of the roles in a software project. ISO/IEC29110 was used to define the roles, along with suggested descriptions of competencies in the development. There are seven roles in the standard, i.e., analyst, customer, designer, programmer, project manager, technical leader, and work team [16]. On the other hand, there are only are three roles in the Scrum methodology—product owner, scrum master, and scrum team. The analysis was conduct by the authors and record results in this section.

Table 4 provides a compliance analysis of the roles performed using the ISO/IEC29110 and Scrum method. The analysis was performed by scoring the compliance of competency from the description and competency of the role. The highest score was three, which indicates that both roles are essentially the same or only slightly different. The lowest was 0, which means that the roles require completely different competencies. The product owner resembles the analyst and customer, based on their function of reacting to the project requirements and recognizing the business purposes of the customer. According to their descriptions, the Scrum master is similar to the technical leader. Through making sure everything goes smoothly and is done in a timely manner, this role is responsible for monitoring the project. The project development team, following the implementation process, is the Scrum team. It was found from this study that the designer, programmer, and work team were the same as the Scrum team. A project manager is a role that is not

identical to the role of a Scrum, as the project management area has to involve managing the resources and making decisions.

**Table 4.** Compliance analysis of roles in ISO/IEC29110 and Scrum methodologies.

| Role | Analyst | Customer | Designer | Programmer | Project Manager | Technical Leader | Work Team |
|---|---|---|---|---|---|---|---|
| Product Owner | 3 | 2 | 0 | 0 | 0 | 0 | 0 |
| Scrum Master | 0 | 0 | 0 | 0 | 0 | 3 | 0 |
| Scrum Team | 1 | 0 | 3 | 3 | 0 | 1 | 2 |

Based on the compliance analysis, the roles used for modeling in EPF were identified to be product owner, Scrum master, and Scrum team, as these showed the highest compliance values. The customer, work team, and project manager were also included the modeling since several of their contributions were not covered by the Scum roles. An example of this is the authorization of user acceptance by the customer or resource management, which is performed in Scrum by the project manager.

The mapping of activities in ISO/IEC29110 and those of the Scrum approach are shown in Tables 5 and 6. The project management process includes four activities of development—project planning, project plan execution, project assessment and control, and project closure. All scrum events can involve the activities of project management. Table 6 is the mapping of software implementation activities and scrum events. Six activities from SI process—SI.1: Software Implementation Initiation, SI.2: Software Requirement Analysis, SI.3: Software Architecture and detailed design, SI.4: Software construction, SI.5: Software integration and tests, and SI.6: Product delivery. All activities were mapped to the Scrum events, except for the sprint retrospective, which had no related activity. Based on the analysis, all of the scrum events involved the ISO/IEC29110 standard. This analysis can be used as a guide for modeling the process in EPF. The breakdown of each activity was assign to the method content, as shown in Section 4. According to the analysis presented in Table 4, the project manager, customer, and work team were not completely defined in the Scrum model. There are slight effects in regard to the model. Some of the tasks were added on to the model in order to fulfill the requirements of the standard. From Table 1, the process performance indicators (outcome) will perform by each role in the team. In the PM process, the project manager performs outcome (b), "the tasks and resources necessary to complete the work shall be estimated (schedule, effort, cost, duration)". Outcome (o), "project closure shall be performed to get the customer acceptance", is performed to by the customer. In the SI process, outcome (k), "consistency and traceability between software requirements, software architectural, software detailed design, and software components shall be established", is achieved by the work team.

**Table 5.** The mapping of activities in the project management (PM) process and Scrum events.

| ISO/IEC29110 Activity | Project Management | | | |
|---|---|---|---|---|
| | Project Planning | Project Plan Execution | Project Assessment and Control | Project Closure |
| Scrum event | | | | |
| Sprint planning | x | | | |
| Daily Scrum | | x | | |
| Sprint | | x | | |
| Sprint Review | | | x | |
| Sprint Retrospective | | | | x |

**Table 6.** The mapping of activities in the software implementation (SI) process and Scrum events.

| ISO/IEC29110 Activity | Software Implementation | | | | | |
|:---:|:---:|:---:|:---:|:---:|:---:|:---:|
| | SI.1 | SI.2 | SI.3 | SI.4 | SI.5 | SI.6 |
| Scrum event | | | | | | |
| Sprint planning | x | x | x | | | |
| Daily Scrum | x | | | | | |
| Sprint | | | x | x | x | |
| Sprint Review | | | | | | x |
| Sprint Retrospective | | | | | | |

Process Modeling on EPF Composer

The EPF composer is a tool for modeling the process of an organization or individual project. The method content is primarily expressed using work products, roles, tasks, and guidance. Guidance can take the form of checklists, examples, or roadmaps. The prepared elements for input to the method content are shown in Figure 6.

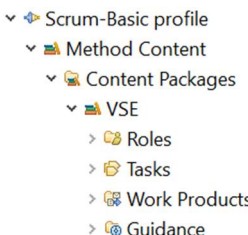

**Figure 6.** Prepared elements for the method content input in EPF Composer.

There are four elements of the prepared content method—roles, task, work product, and guidance. Based on the mapping of the roles outlined in ISO/IEC29110, the Scrum model was used to define the required roles for the modeling—namely, customer, designer, product owner, programmer, project manager, and scrum master. All of the roles were created in the category of the role in the VSE Composer, as shown in Figure 7. A task is an activity in the process, presented along with the input and output of the activity. Each task contains the description, step, associated role, and work product. Examples of tasks are shown in Figure 8.

A work product is an item associated with a task, which acts as the input or the output of the task. As shown in Figure 9, there were 15 work products selected from ISO/IEC29110 and the Scrum methodology. The attribute of the work product is "state", the status of the work product. Examples of these states are draft, reviewed, baseline, and release.

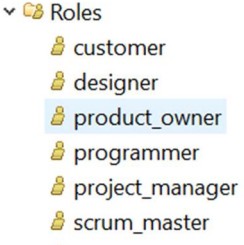

**Figure 7.** Defined roles in the modeling.

```
∨ 📁 Tasks
    ▷ pm.1_project_planning
    ▷ pm.1.1_get_the_approvement_from_customer
    ▷ pm.1.2_define_delivery_instruction
    ▷ pm.1.3_identify_the_specific_task_and_resource
    ▷ pm.1.4_prepare_the_development_enviroment
    ▷ pm.1.5_release_planning
    ▷ pm.2_project_plan_execution
    ▷ pm.2.1_describes_the_feature
    ▷ pm.2.2_define_sprint_goal
    ▷ pm.2.3_negotiation
    ▷ pm.2.4_daily_scrum_meeting
    ▷ PM.2.5 report the progress to CUS
    ▷ pm.3_project_assessment_and_control
    ▷ pm.3.1_sprint_review_meeting
    ▷ pm.3.3_risk_identification
    ▷ pm.4_project_closure
```

**Figure 8.** Examples of tasks in VSE Composer.

```
∨ 🗂 Work Products
    📄 minute_of_meeting
    📄 product_backlog
    📄 project_plan
    📄 project_repository
    📄 release_burndown_chart
    📄 software
    📄 software_component
    📄 software_design
    📄 sprint_backlog
    📄 sprint_burndown_chart
    📄 statement_of_work
    📄 task_board
    📄 testing_document
    📄 traceability_record
    📄 user_acceptance_record
```

**Figure 9.** The selected work products from ISO/IEC29110 and Scrum.

Guidance is supporting information for the other elements in the content method. There are many types of guidance that can define the content, such as checklists, guidelines, roadmaps, concepts, and reports. In this modeling, checklist was selected as the process outcomes from Table 1. The checklists are the requirement to achieve the standard. The Scrum activity consists of the daily Scrum, sprint planning, sprint retrospective, and sprint. These components were defined in guidelines in the guidance section of the VSE method content.

*3.3. Process Implementation on EPF*

Capability patterns and delivery processes are two types of process. Delivery processes represent a complete and integrated process template for carrying out a specific type of project. This process describes a complete end-to-end project life-cycle and is used as a reference for running projects with similar characteristics. Capability patterns have the process that express and communicate process knowledge relating to a key area of interest, such as a discipline or best practice. They are also used as building blocks to assemble delivery processes or larger capability patterns. In the design of the process in our study,

the capability pattern was assigned based on the process and activity of the ISO/IEC29110 standard [17]. Figure 10 shows the designed structure of the capability pattern in the EPF.

> ∨ 🔹 Processes
>  ∨ 🔹 Capability Patterns
>    ∨ 🔹 PM
>       🔹 PM.1Project planning
>       🔹 PM.2 Project plan execution
>       🔹 PM.3Project Assessment and Control
>       🔹 PM.4Project closure
>    ∨ 🔹 SI
>       🔹 SI.1Software Implementation Initiation
>       🔹 SI.2Software Requirements Analysis
>       🔹 SI.3Software Architectural and Detailed Design
>       🔹 SI.4 Software Construction
>       🔹 SI.5 Software Integration and Tests
>       🔹 SI.6 Product Delivery

**Figure 10.** Structure of capability patterns in the EPF.

The capability pattern of the process design consists of two main process packages that follow the ISO/IEC29110 process—the PM and SI packages. Each package includes the capability pattern as an activity of the process. The individual capability pattern includes the content defined in the method content—roles, tasks, work product, checklist, and guidance. In Figure 11 shows the elements to construct a capability pattern. The IDEF0 (Definition of Integration Modeling Function) conceptual method was applied in the capability pattern design. IDEF0 is a simple process design method. There are five elements to considers in the process design, which are input, output, task, role, and constraint [18,19]. The input, output, and role were selected from the package of work product and role. The checklists and guidance are assigned as a constraint of the process. The designed task is assigned by following the naming approach.

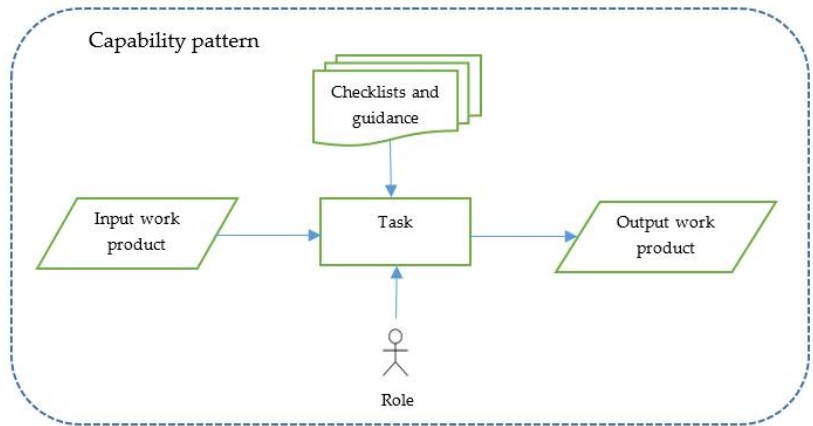

**Figure 11.** The structure of capability pattern.

[process name].[activity number].[sub-number] [task name].

### 3.4. Example of Process Visualization in the EPF

Based on the analyzed information of ISO/IEC29110 and the Scrum methodology, the processes can be visualized on the EPF based on their capability patterns. This section presents an example of the implementation of this in relation to the project management process. The project management process consists of four capability patterns. The capability pattern of PM begins with—PM.1, project planning; PM.2, project plan execution; PM.3,

project assessment and control; and PM.4, project closure. The capability pattern is designed to assign these phases according to the work breakdown structure. The EPF allowed for the construction of the process using the content package from the method content. An example of an implemented capability pattern is presented in the following section.

- PM.1, Project planning

Project planning is an activity involving the planning of all resources for project development. There are three phases in this activity. Inception is the phase of gathering the agreement from the customer. Planning is the phase of planning all the required resources in the project, e.g., people, time, tools and equipment, budget, and methodology. The last phase is releasing the plan, in which the manager informs all of the stakeholders in the development of the project plan.

As shown in Figure 12, PM.1 includes three phases. Each activity consists of activity and a task. The inception phase is executed by reviewing the statement of work activity. A task required to complete this activity is PM.1.1, i.e., getting the approval from the customer. The detailed activity diagram for the inception phase is shown in Figure 13. This figure shows the detailed activity diagram involving the project manager, who executes task PM.1.1 using the work product name and the statement of work as the input. The output of PM1.1 is the same work product, but its status is changed.

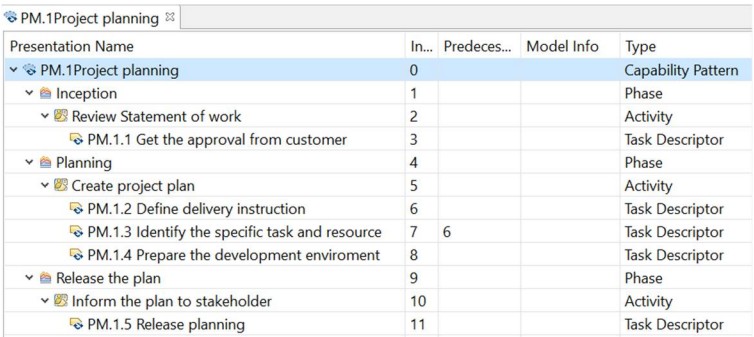

| Presentation Name | In... | Predeces... | Model Info | Type |
|---|---|---|---|---|
| ⌄ PM.1Project planning | 0 | | | Capability Pattern |
| ⌄ Inception | 1 | | | Phase |
| ⌄ Review Statement of work | 2 | | | Activity |
| PM.1.1 Get the approval from customer | 3 | | | Task Descriptor |
| ⌄ Planning | 4 | | | Phase |
| ⌄ Create project plan | 5 | | | Activity |
| PM.1.2 Define delivery instruction | 6 | | | Task Descriptor |
| PM.1.3 Identify the specific task and resource | 7 | 6 | | Task Descriptor |
| PM.1.4 Prepare the development enviroment | 8 | | | Task Descriptor |
| ⌄ Release the plan | 9 | | | Phase |
| ⌄ Inform the plan to stakeholder | 10 | | | Activity |
| PM.1.5 Release planning | 11 | | | Task Descriptor |

**Figure 12.** PM.1 (project planning) capability pattern.

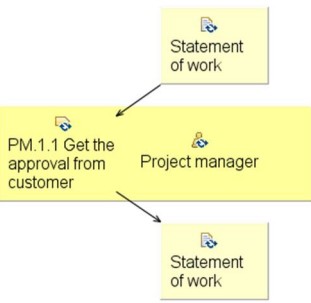

**Figure 13.** The detailed activity diagram for the PM.1.1 task.

The task description of PM1.1 is shown in Figure 14. It defines the related elements of this task, including discipline, role, work product (input and output), process usage, step, and checklist. The project manager is the primary performer who executes this task, whereas the customer is an additional performer who also contributes to the accomplishment of the task. The statement of work is a work product, representing the input and the output. The project management process, project planning, inception, and review statement of work use the process usage, presenting the PM1.1. There are two steps in PM1.1, i.e., communicating with the customer and gaining their approval (sign-off). The checklist is the achieved outcome of the task. This task gives the outcome (a) in the PM process. The second phase is planning. There are three tasks in this phase; PM.1.2, PM.1.3, and PM.1.4. The activity diagram is presented in Figure 15.

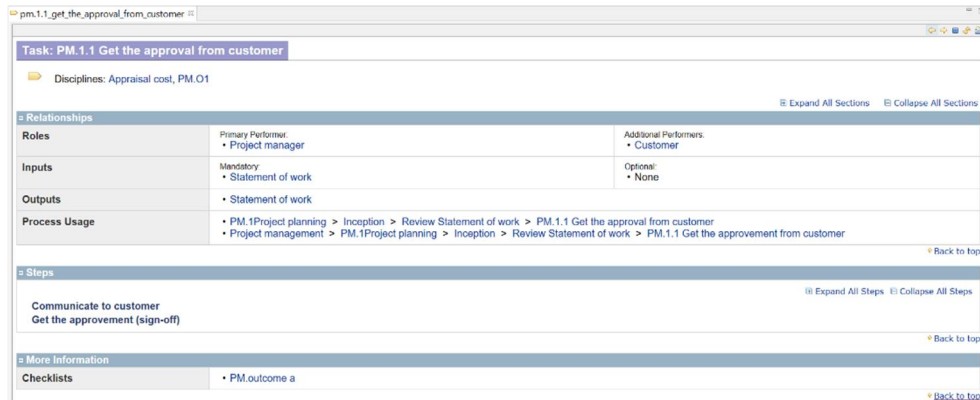

**Figure 14.** The task description of PM1.1.

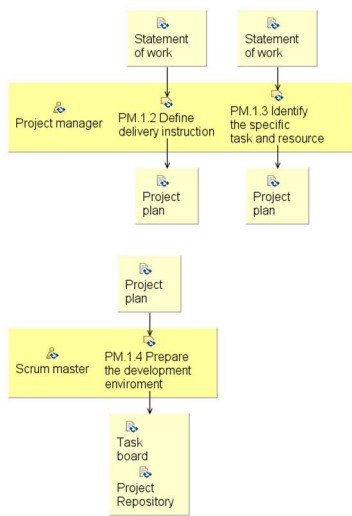

**Figure 15.** The detailed activity diagram of the planning process.

The project manager and the Scrum master are the key roles in executing tasks. The project manager has been involved in tasks PM1.2 and PM1.3 The statement of work is an input work product, and the project plan is an output. The Scrum master responds to task PM.1.4 and creates the task board and the project repository, using the project plan from the project manager. The last phase of the PM.1 capability pattern is to release the planning. This phase consists of one task, namely, PM.1.5. The detailed activity diagram for this phrase is shown in Figure 16. There are four capability patterns in the PM process and six capability patterns in the SI process, which were implemented in EPF Composer. All of the elements from the content method were integrated into each capability pattern which was task, role, work product, and guidance (checklist).

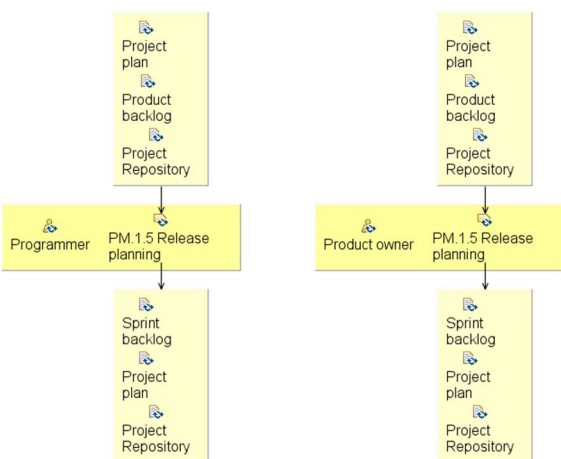

**Figure 16.** The detailed activity diagram of phrase PM.1.5.

## 4. Discussion

This paper presents the implementation of process visualization using the EPF Composer tool. The EPF composer allows an engineer to create and modify the process from an authoring perspective. The designed process was created using the library in the Composer program. The method content package comprises the main data, using work products, roles, tasks, and guidance.

It is possible to publish the output in the form of a web application that is convenient to use in an organization. In our case study, the ISO/IEC29110 software development standard in a basic profile and the Scrum methodology were the implementation sources. The result of the experiment was a framework based on the analysis of the ISO/IEC 29110 basic profile and Scrum. There were a few elements of the Scrum methodology that could not be mapped to the requirements of the standard. To meet the requirements of the standard, the author must add these elements to the model. The framework is a structure based on the content method developed in EPF Composer. There were six roles, fifteen work products, thirty-two guidance, and thirty-one tasks in the PM and SI processes. Comparison number of tasks, the new framework can reduce the number of tasks in the ISO/IEC29110 standard from sixty-six tasks to thirty-one ($-53\%$). The activities from each process were applied using a capability pattern in the process design. All of the data in the implemented framework can be generated to a web application which includes the description, activity diagram, and the related data of the method content. This framework can act as a mock-up of the initial process for VSEs that are interested in applying the ISO/IEC29110 standard and Scrum methodology in a software project.

In the case study, the framework was applied to the VSE who interests in getting the certification of ISO/IEC29110 basic profile. The organization size is ten employees. There are several problems in the project development. e.g., miscommunication in team, task management, project delay, and product quality control. The VSE uses the framework as a quality plan for their projects. The manager adjusts the framework in the EPF tool for more convenience in the project. e.g., add the special role (chief executive officer and sale), add the new software tool for management. The three-month software project was assigned as a pilot project of this study. In the project development, the manager can monitor the quality of development by using the checklists of each capability pattern. The task descriptions and activity diagrams on a webpage can use to describe a process to the team. More convenience assigns the task to team members, and all of the artifacts are arranged for a certification audit. From the feedback from this case study, the VSE can solve the problems and satisfy their requirements by using the framework without an external consultant and achieve ISO/IEC29110 certification.

There are many effective software development methodologies from people in the industry that need to transfer. e.g., DevOps, multiple scrums, hybrid software process [20].

The methodologies from this article allow extending their methods or principles in the future. The improved process or the framework can transfer to the VSE to satisfy their business and customer requirement.

## 5. Conclusions

This article focuses on implementing the process of software development with reference to the ISO/IEC 29110 standard and the Scrum methodology. The EPF Composer program was capable of designing a new framework for the software development process. The created libraries and web applications were the output of its implementation. VSEs can use this library to improve their software development processes. The VSE can view the details of tasks explicitly. All of the participants in software development can understand their own tasks and their work products by following the reference processes outlined in the EPF. The tool allows the user to adjust the detail of the process to optimize development. On the other hand, VSEs can use this library in preparation for certifying the ISO/IEC 29110 standard. An auditor can use it to inspect the process of development and view the workflow within an organization. The strengths and weaknesses of the organization are thus easier to define for corrective action and improvement.

**Author Contributions:** Conceptualization, K.S. and S.R.; methodology, K.S. and S.R.; software, K.S.; validation, K.S. and S.R.; formal analysis, K.S. and S.R.; writing—original draft preparation, K.S.; writing—review and editing, K.S. and S.R.; visualization, K.S. and S.R. Both authors have read and agreed to the published version of the manuscript.

**Funding:** This research received funding from the Graduate school, Chiangmai University.

**Conflicts of Interest:** The authors declare no conflict of interest.

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
