# Peer review of "The Visualization of ISO/IEC29110 on SCRUM under EPF Composer"

_information, doi:10.3390/info12050190_

Round 1
Reviewer 1 Report
The paper is quite interesting, well written, and easy to follow.
However, what refrain me from accepting the work is its scientific contribution: To better highlight, the contribution authors should:
- add a crisper discussion on related works discussing the novelty of their approach
- discuss the research field
Author Response
Point 1:add a crisper discussion on related works discussing the novelty of their approach
Response 1: More discussion and the case study and result are described in part 4. one of the VSE applied the framework to their organization. The framework can help this VSE to satisfy the requirements.
Point 2 :discuss the research field.
Response 2: Add more literature review in part 1. The problems are mentioned in the statement. Kuhrmann et al. “Software and system development in practice: Waterfall, Scrum, and beyond” this article shows the problem of the standard and development model implementation.
Best Regards,
Reviewer 2 Report
The paper describes a tool to enable ISO 29110-compliant agile (scrum) software development life cycles.
The paper is written in good English and rather easy to understand. I think it addresses a real problem in software development that is worthwhile to investigate further.
However, the paper appear do not this. Instead, it seems to provide too many details and examples, while not describing the actual approach in enough depth. I am missing a clear description of the actual scientific contributions of the paper. It describes examples of how a Scrum process can be modeled inside the Eclipse Process Framework Composer. My main concern is that the paper does not make it clearer how this does help VSEs.
Abstract:
The paper states that there are strong incompatibilities between standards and the agile approach. However, although plausible, this discrepancy is not substantiated or further discussed in the rest of the paper.
Section 1:
I agree that the process requirements that help achieving high quality software are hardly compatible with agile software development; see, e.g., Prause et al. (Software product assurance at the German space agency, 2016): “Agile software development might not be suitable for flight software development. […] trying to squeeze agile development into the existing, more traditional framework of development. Experts have made the mention of the infamous agile waterfall, i.e., first several iterations of requirements engineering, then several iterations of design, then several iterations of coding and, finally, several iterations of testing.” You might also want to take a look at the publications of the HELENA project on hybrid development methods (e.g., Kuhrmann et al. “Software and system development in practice: Waterfall, Scrum, and beyond”, “Hybrid Software Development Approaches in Practice: A European Perspective”, or others).
The contributions of this paper are not very clear. The last paragraph of Section 1 states that several cases of implementing ISO29110 in an agile development are studied. However, I do not find such cases in the paper. I have a feeling that this refers to the modeling examples in Section 3. But these are no real cases from the real-world, are they? Please make it more clear, what your actual contributions are. What will you show in this paper?
Please a short paragraph that gives an overview of the story you tell, and how this story flows through all of the sections of this paper.
Section 2:
Should probably be named “Backgrounds” or “Related work”. I do not see that what you present here are Materials or Methods as you would expect from the IMRaD methodology. It is ok to describe your perspective on Scrum but this is rather background knowledge.
You do not describe any Methods. The method is usually what results in the results that you present in the next section.
Please add a short introduction to Sections 2.1.1 and 2.1.2. Do not just present bullet points.
Figure 1: Just for optical reasons, you might consider to put the URL from the Figure caption into the references.
Section 2.4: Remove the year from Buchalcevova. You have [9] at the end of the sentence.
Section 3:
Missing introductory sentence.
I like the comparison of role descriptions from ISO29110 and Scrum. However, a more clear description of how the results were obtained is missing. What was the exact approach? Who did the comparison? Did the authors (or just one of them?) read the role descriptions in both standards (?) and then note what the understand? A more rigorous description of the Material and Methods is needed.
What do the letters after “outcome (a/o/k)” mean?
The section presents to many screenshots where it is unclear what their actual benefit for the reader is. It is probably too much details and too few actual concepts presented here. Do these screenshots just document how you actually implemented ISO29110 and Scrum in EPF?
Section 4:
The paper mentions that a case-study was done. I did not find a case study in this paper. Please refer to standard literature like Runeson et al. (Case study research in software engineering: Guidelines and examples) on how to conduct case studies.
The paper mentions that an experiment was done. I did not find an experiment in this paper. You would describe an experiment using the IMRaD structure. But there was no experiment. I just found an implementation of ISO29110 and Scrum in the EPF tool. The only thing that appears to be remotely empirical is the comparison of roles in Section 3. Empirical work usually should be accompanied by a “threats to validity” Section.
Author Response
the response is in the attachment.
Best Regards,

Reviewer 3 Report
In this manuscript, the authors propose an implementation of visualization of software development process by using Composer tool in Eclipse Process Framework (EPF).
The manuscript is not well-structured, for example, section “Related work” is missing.
In the “Introduction” section, the research goal, novelty and contributions are not clearly formulated. For example, the last edition of ISO/IEC TR 29110-1:2016 is from 2016 and it is unclear what is the author motivation to study this standard five years later.
Please, add some details about similar previous studies by using references from the last five years.
The source code is missing.
Future plans are also missing.
Some technical remarks
Table 2: Please add some details about PM.O1 and Trello (last column).
l. 8: “In the SI process,…” – “In the software implementation (SI) process,…”. Remove the definition of SI from the caption of Table 5.
l. 13: “which was task, role, work product, guidance, and checklist” -> “which was task, role, work product, guidance (checklist)”.
I am sorry to reject the manuscript.
Author Response
the response is in the attachment.
Best regards,

Round 2
Reviewer 1 Report
Authors addressed the comments. The paper can be accepted.
Author Response
Dear reviewer,
the manuscript improvement.
point 1: English language and style are fine/minor spell check required
response 1: the English language has been re-check spelling and grammar.
Thank you for the comments and accepts our work.
Best Regards,
Kittitouch s.
Sakgasit r.
Reviewer 3 Report
The Visualization of ISO/IEC29110 on SCRUM under EPF Composer
The quality of revised information-1166542 “The Visualization of ISO/IEC29110 on SCRUM under EPF Composer” has been improved and the manuscript now meets the requirements of MDPI Information Journal.
My recommendation is “Accept as is”.
Some technical remarks:
p. 4: "thirteen outcomes" - Please, check the outcomes list. (j) is missing.
p. 10: "outcome (a), “the tasks and resources necessary..." -> "outcome (b), “the tasks and resources necessary..."
p. 17: "Future plans, there are many effective software..." - Please, edit this sentence.
Author Response
Dear reviewer,
We agree to edit the manuscript. Our improvement as following.
point1 p. 4: "thirteen outcomes" - Please, check the outcomes list. (j) is missing.
response 1: Move the letter (j) to the newline >>
j) software shall be tested and verified, the results shall be recorded;
p. 10: "outcome (a), “the tasks and resources necessary..." -> "outcome (b), “the tasks and resources necessary..."
response 2: Change letter a to b.
p. 17: "Future plans, there are many effective software..." - Please, edit this sentence.
response 3: Edit the sentence.
There are many effective software development methodologies from people in the industry that need to transfer. e.g., DevOps, multiple scrums, hybrid software process [20]. The methodologies from this article allow extending their methods or principles in the future. The improved process or the framework can transfer to the VSE to satisfy their business and customer requirement.
point 4: English language and style are fine/minor spell check required
response 4: the English language has been re-check spelling and grammar.
Thank you for the comments and accepts our work.
Best Regards,
Kittitouch s.
Sakgasit r.